# Contrasting roles for IKK-regulated inflammatory signalling pathways for development and maintenance of type 1 and adaptive γδ T cells

**Farjana Islam[†], Cayman Williams, Thea Hogan, Louise V Webb[‡], Ines Boal-Carvalho*[§], Benedict Seddon***

Institute of Immunity and Transplantation, Division of Infection and Immunity, University College London, London, United Kingdom

**\*For correspondence:**
icarvalho@virtus-rr.com (IB-C);
benedict.seddon@ucl.ac.uk (BS)

**Present address:** [†]Department of Biochemistry and Molecular Biology, Shahjalal University of Science and Technology, Sylhet, Bangladesh; [‡]Autolus Therapeutics plc, London, United Kingdom; [§]Virtus Respiratory Research Ltd, London, United Kingdom

**Competing interest:** The authors declare that no competing interests exist.

## eLife Assessment

This study reports **important** findings regarding the role of the NF-kB signaling pathway in the development and long-term survival of gamma delta T cells. The authors report disparate roles of IKK-dependent NF-kB activation in the development and long-term survival of gamma delta T cell subsets. The approach and methodology employed is **convincing**. This work will be of great interest to immunologists interested in innate-like T cell biology and in T cell development.

**Abstract** The inhibitor of kappa-B kinase (IKK) complex is a critical regulator of cell death and inflammatory signalling in multiple cell types. Phosphorylation of IκB proteins by IKK results in their degradation and consequent activation of NF-κB transcription factors. RIPK1, a critical cell death regulator, is also a direct target of IKK kinase activity, thereby repressing its cell death activity. In αβ T cells, the RIPK1 kinase activity of IKK is critical for normal thymic development while mature αβ T cells require IKK for both activation of NF-κB dependent survival programmes and repression of RIPK1. γδ T cells play a unique and versatile role in host immunity with specific effector functions that enable them to act as early responders in immune defence. The role of IKK-regulated pathways in their development and survival is not known. Here, we dissect the function of IKK and downstream pathways for normal γδ T cell homeostasis. We find that IKK is critical to establish replete γδ T cell populations, but that mechanism varys between different subsets. Type 1 γδ T cells require IKK-dependent NF-κB activation for their generation, while IKK is redundant for development of adaptive γδ T cells. Instead, IKK-dependent NF-κB activation is required for their long-term survival. We also find evidence that IKK repression of RIPK1 is required for survival of peripheral but not thymic γδ T cells. Ablation of CASPASE8 did not rescue γδ T cells in the absence of IKK but rather revealed a potent sensitivity of all γδ subsets to necroptosis, which was rescued by kinase-dead RIPK1. Overall, we reveal critical requirements for IKK-regulated inflammatory pathways by γδ T cells that contrast with those of αβ T cells, and between different subsets, highlighting the complexity of the regulation of these pathways in the adaptive immune system.

## Introduction

The NF-κB family of transcription factors plays critical roles in controlling development and function of many cell types (*Bonizzi and Karin, 2004*). Canonical NF-κB signalling is mediated by hetero- or homodimers of p50, RELA, and cREL family members that are sequestered in the cytoplasm by

inhibitory proteins, the inhibitors of kappa B (IκB) family and the related protein NFκB1. The key regulator of NF-κB dimer release is the inhibitor of kappa-B kinase (IKK) complex, a trimeric complex of two kinases, IKK1 (IKKα) and IKK2 (IKKβ), and a third regulatory component, NEMO (IKKγ). IKK phosphorylates IκB proteins, targeting them for degradation by the proteasome and releasing NF-κB dimers to enter the nucleus.

In αβ T cells, activation of NF-κB is a well-recognised critical early event during T cell receptor antigen recognition (*Gerondakis and Siebenlist, 2010*). In the absence of REL subunits, or upstream NF-κB activators, such as TAK1 or IKK complex, T cells fail to blast transform or enter cell cycle (*Webb et al., 2019*; *Xing et al., 2016*). Consequently, mice with T cell-specific ablation of RELA and/or cREL have peripheral naive T cells, but lack effector or memory phenotype T cells (*Webb et al., 2019*; *Zheng et al., 2003*). As well as regulating activation, the NF-κB signalling pathway also has important functions in the development and survival of αβ T cells. While activation of NF-κB by TCR appears redundant for normal selection of thymocytes during thymic development (*Schmidt-Supprian et al., 2004*; *Webb et al., 2019*), signals from TNF and other TNF receptor superfamily members (TNFRSF) are important for survival and differentiation of post-selection single positive thymocytes. NF-κB is required for re-expression of *Il7r* following thymic development, that is necessary for long-term IL-7 dependent survival of peripheral naive T cells (*Miller et al., 2014*; *Silva et al., 2014*). Additionally, long-term survival of fully mature naive CD4[+] T cells is dependent upon tonic NF-κB since CD4[CreERT]-induced deletion of REL subunits results in a substantial loss of T cells (*Carty et al., 2023*).

In addition to activating NF-κB dependent transcriptional activity to regulate T cell survival, TNF-induced NF-κB signalling pathways also control cell death through the activity of IKK. In addition to its function as an IKK, the IKK complex also directly controls cell survival independent of NF-κB activation. While thymic development is largely normal in the absence of REL subunits (*Webb et al., 2019*), IKK deficiency in αβ T cells results in a profound block in thymic development that is arrested at the SP stage (*Schmidt-Supprian et al., 2003*) due to TNF-induced apoptosis (*Webb et al., 2016*). Ligation of TNFR1 causes recruitment of TRADD, TRAF2, and the serine/threonine kinase RIPK1. The ubiquitin ligases TRAF2, cellular inhibitor of apoptosis proteins (cIAPs), and the linear ubiquitin chain assembly complex (LUBAC), add ubiquitin chain modifications to themselves and RIPK1, creating a scaffold that allows recruitment and activation of the TAB/TAK and IKK complexes that in turn activate NF-κB. This is termed complex I (*Annibaldi and Meier, 2018*; *Vandenabeele et al., 2010*). A failure to maintain the stability of this complex results in the formation of cell death inducing complexes composed of TRADD, FADD, CASPASE8, and RIPK1 that induce apoptosis, a function dependent upon RIPK1 kinase activity (*Annibaldi and Meier, 2018*; *Dondelinger et al., 2016*; *Ting and Bertrand, 2016*). Phosphorylation of RIPK1 by IKK blocks RIPK1 kinase activity and therefore its capacity to induce apoptosis (*Dondelinger et al., 2015*). In thymocytes, it is this function of IKK, and not NF-κB activation, that is critical for their survival and onward development and accounts for the phenotype observed in IKK deficiency (*Blanchett et al., 2022*; *Webb et al., 2019*). Survival of mature T cells depends upon both the IKK capacity to repress RIPK1 and its function to activate NF-κB (*Carty et al., 2023*). RIPK1-dependent cell death of thymocytes and mature naive T cells is mediated by apoptosis, rather than necroptosis, since cell death is CASPASE8 dependent (*Carty et al., 2023*). Thymocytes are not susceptible to necroptosis as they lack expression of MLKL (*Webb et al., 2019*), a key effector molecule for mediating necroptotic cell death. Similarly, resting peripheral αβ T cells are resistant to necroptotic cell death, since *Casp8* deletion has little impact on peripheral αβ T cell compartments (*Ch'en et al., 2008*; *Ch'en et al., 2011*). In contrast, activated T cells are acutely susceptible to necroptosis and readily undergo necroptotic cell death in the absence of *Casp8* expression during LCMV infection or following activation in vitro (*Ch'en et al., 2008*; *Ch'en et al., 2011*).

γδ T cells are a distinct subset of T lymphocytes that play a unique and versatile role in host immunity. While conventional αβ T cells rely on MHC-restricted antigen presentation, γδ T cells recognise stress-induced ligands, phosphoantigens, and non-peptidic molecules directly (*Deseke and Prinz, 2020*). Distinct subsets of innate-like γδ T cells develop with specific effector functions, such as type 1 and 17 γδ T cells (*Muñoz-Ruiz et al., 2017*; *Vantourout and Hayday, 2013*). Weak TCR signalling during thymic development is associated with generation of type 17 γδ T cells, which are CD44[hi] and lack CD27 expression, and develop as a finite wave in a very specific window of embryogenesis (*Muñoz-Ruiz et al., 2017*). In contrast, type 1 γδ T cell development is associated with strong TCR signalling and cells assume a CD122[hi]CD27[+] phenotype during thymic development. The remainder

of γδ T cells in lymphoid tissues are undifferentiated and exhibit a more naive CD44$^{lo}$CD27$^+$ phenotype and are termed by some as naive or adaptive γδ T cells. The pre-differentiated type 1 and 17 states enable them to act as early responders in immune defence. As such, while it seems likely that inflammatory NF-κB and cell death signalling pathways would be important regulators of γδ T cell development and homeostasis, as is the case for αβ T cells, very little is currently known. Evidence that NF-κB signalling could be important comes from the observation that the TNFRSF member CD27 is an important regulator of type 1 γδ T cell development (*Ribot et al., 2009*) and is also an activator of NF-κB signalling required for survival of αβ T cells (*Silva et al., 2014*). However, whether and how inflammatory signalling pathways mediated by IKK and NF-κB for the normal homeostasis of γδ T cells remains unknown.

In the present study, we used mouse genetics to investigate the role of IKK and NF-κB survival and cell death pathways for homeostasis of γδ T cells. We found evidence that thymic development of type I γδ T cells was dependent on these pathways, but redundant for adaptive γδ T cells. However, long-term survival of adaptive γδ T cells was dependent upon both NF-κB and, in part, RIPK1-dependent survival pathways and both populations were highly susceptible to necroptotic cell death.

## Results
## IKK signalling is required for generation of type 1 γδ T cells and maintenance of adaptive γδ T cells

We first asked whether IKK signalling was required for γδ T cell specification and development of naive/adaptive (adaptive hereon) and type 1 γδ T cells in the thymus of adult mice lacking expression of IKK proteins in the T cell compartment. Type 17 γδ T cells develop as a finite wave in a very specific window of embryogenesis (*Muñoz-Ruiz et al., 2017*). Since our analysis was focused on adult genetic mutants, we only assessed type 17 γδ T cells in secondary lymphoid organs. We generated mice with conditional genes encoding IKK1 and IKK2 protein (*Chuk$^{fx}$* and *Ikbkb$^{fx}$*, respectively) and an iCre transgenic construct controlled by huCD2 expression elements (IKKΔT$^{CD2}$ mice hereon). huCD2$^{iCre}$ is expressed in CLP in bone marrow, and Cre-mediated gene deletion is evident almost all of the earliest thymic progenitors that enter the thymus (*Siegemund et al., 2015*), so it is ideal to target gene deletion prior to γδ T cell specification that occurs during DN2 (*Fiala et al., 2020*). In confirmation, we analysed huCD2$^{iCre}$-mediated Rosa26RmTom *Cre* reporter expression in adaptive, type 1, and type 17 γδ T cell subsets and found ubiquitous expression of reporter in all subsets (*Figure 1—figure supplement 1*). Analysing thymi from IKKΔT$^{CD2}$ mice revealed normal representation and numbers of both CD25$^+$ CD27$^+$ TCRδ$^+$ progenitor cells and total CD25$^-$ CD27$^+$ TCRδ$^+$ T cells, suggesting that IKK signalling was not required for either specification of the γδ T cell lineage from uncommitted DN precursors or subsequent thymic development of γδ T cells. HSA expression was high in both populations in IKKΔT$^{CD2}$ mice, confirming their immature status (*Figure 1A*). Type 1 γδ T cells can be identified by their expression of CD122. Analysing subsets of CD25$^-$ CD27$^+$ TCRδ$^+$ T cells revealed that CD122$^-$ adaptive γδ T cells were present in normal numbers while numbers of CD122$^+$ type 1 γδ T cells were significantly and substantially reduced in the absence of IKK activity (*Figure 1B*).

The thymic phenotype of IKKΔT$^{CD2}$ mice suggested that IKK signalling is required for the development of type 1 γδ T cells. Analysing numbers of type 1 γδ T cells recovered from lymph node (LN) and spleen confirmed this view since IKKΔT$^{CD2}$ mice were almost completely devoid of this subset (*Figure 1C, D*). Strikingly, adaptive γδ T cells were also largely absent from secondary lymphoid organs, both in terms of their frequencies amongst total lymphocytes (*Figure 1D*, *Figure 1—figure supplement 2*) and in terms of absolute numbers of cell recovered from LNs and spleen (*Figure 1D*). Only a small population of CD27$^-$ type 17 γδ T cells was detectable in the periphery of IKKΔT$^{CD2}$ mice, but these were also significantly reduced in number compared to Cre –ve littermates (*Figure 1D*). Together, these data suggest that IKK signalling is required for development of type 1 γδ T cells and for the long-term survival of adaptive γδ T cells.

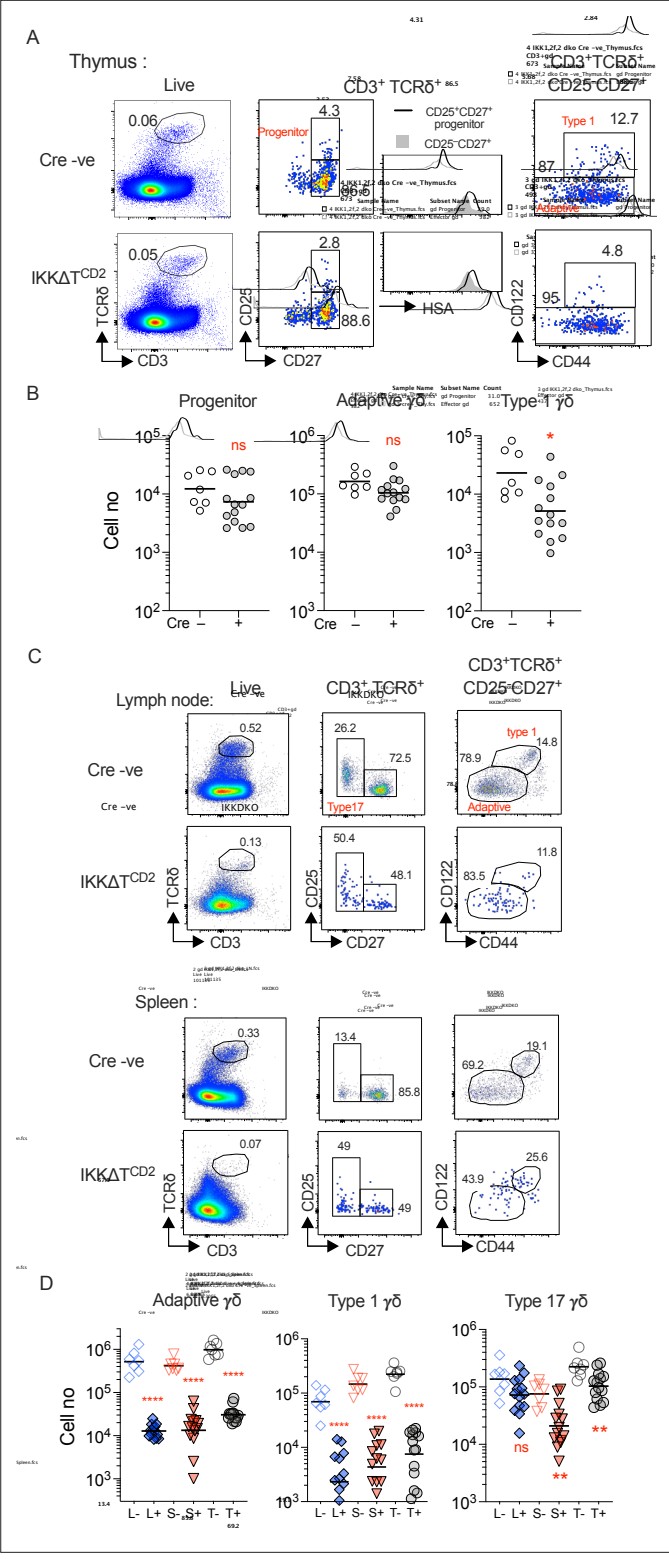

**Figure 1.** Development of type 1 and persistence of adaptive and type 17 γδ T cells depends on IKK expression. Thymi, lymph nodes, and spleen from IKKΔT$^{CD2}$ mice ($n$ = 14) and Cre −ve littermates ($n$ = 7) were enumerated and analysed by flow. (**A**) Representative flow plots are of thymocytes with the indicated gates, showing gates used to identify progenitor, adaptive, and type 1 subsets intrathymically (see Materials and methods for details). (**B**) Scatter plots are of total cell numbers of the indicated subset from the thymus of IKKΔT$^{CD2}$ mice or Cre −ve littermates. Horizontal bars are mean. (**C**) Representative flow plots illustrate the gating strategy to identify adaptive, type

*Figure 1 continued on next page*

*Figure 1 continued*

1, and type 17 subsets in lymph nodes and spleen (see Materials and methods for details). (**D**) Scatter plots are of total cell numbers recovered from lymph nodes (L), spleen (S), and both spleen and lymph nodes combined (T) of the indicated subset from IKKΔT$^{CD2}$ mice (+) or Cre −ve littermates (−). Horizontal bars are mean. Fractional representation of different γδ subsets in different organs is shown in *Figure 1—figure supplement 2A*. Data are pooled from multiple batches of mice analysed. Horizontal lines indicate mean. *p < 0.05, **p < 0.01, ****p < 0.0001, Mann–Whitney test.

The online version of this article includes the following source data and figure supplement(s) for figure 1:

**Source data 1.** Cell enumerations of different gamma-delta subsets in different lymphoid organs.

**Figure supplement 1.** huCD2$^{iCre}$ targets Cre activity to γδ T cells in the thymus.

**Figure supplement 2.** Representation of different γδ T cell sub-populations in different mouse strains.

## Additional *Casp8* deficiency rescues generation type 1 γδ development in the thymus but not maintenance of peripheral γδ T cells in the absence of IKK

In αβ T cells, IKK signalling is required to repress acute induction of *Casp8*-dependent cell death in thymocytes and peripheral T cells. In thymocytes, this survival function is entirely independent of NF-κB, while in mature peripheral T cells, IKK also triggers NF-κB transcriptional activity that contributes to their survival (*Carty et al., 2023*). To test which functions of IKK are required by different γδ T cell populations, we first analysed IKKΔT$^{CD2}$ mice with additional deletion of *Casp8* (Casp8.IKKΔT$^{CD2}$ mice). In the thymus, numbers of progenitor γδ T cells in Casp8.IKKΔT$^{CD2}$ mice were similar to those of Cre −ve littermates (*Figure 2A*), as observed in IKKΔT$^{CD2}$ mice. In contrast, numbers of adaptive γδ T cells in Casp8.IKKΔT$^{CD2}$ mice exhibited a modest but statistically significant reduction, while numbers of type 1 γδ T cells appeared to be rescued to levels similar to Cre −ve controls, suggesting that development of this subset was *Casp8* dependent in the absence of IKK expression. However, this rescue appeared limited in nature, since analysing γδ T cells in the periphery of Casp8.IKKΔT$^{CD2}$ mice failed to reveal any detectable rescue of cell numbers of any γδ T cell subsets in the LN and spleen (*Figure 2B, C*).

## Partial rescue of IKK-deficient peripheral γδ T cells by kinase-dead RIPK1

In αβ T cells, IKK kinase activity blocks CASPASE8-dependent apoptosis by directly phosphorylating RIPK1, which acts to repress its kinase activity that is required to induce CASPASE8-dependent cell death (*Blanchett et al., 2022*; *Webb et al., 2019*). The failure of CASPASE8 ablation to rescue peripheral γδ T cells potentially implied that IKK was not required to repress extrinsic cell death pathways in γδ T cells. However, while *Casp8* deletion would serve to protect cells from CASPASE8-dependent apoptosis, it could also result in triggering death instead by necroptosis. In T cells, extrinsic death pathways triggering CASPASE8-dependent apoptosis or necroptosis both depend on the kinase activity of RIPK1. Therefore, to test whether *Casp8* deletion might instead be triggering necroptosis in Casp8.IKKΔT$^{CD2}$ mice, we analysed peripheral γδ T cell numbers in IKKΔT$^{CD2}$ mice expressing kinase-dead RIPK1$^{D138N}$. As described earlier, IKKΔT$^{CD2}$ mice are almost completely devoid of γδ T cells, except for type 17 cells that were present, albeit in reduced numbers (*Figure 3A, B*). Analysing IKKΔT$^{CD2}$ RIPK1$^{D138N}$ mice revealed a small but significant population of CD27$^+$ γδ T cells in the periphery of mice, that included both adaptive and type 1 subsets (*Figure 3A*). In contrast, the reduction in type 17 γδ T cells observed in IKKΔT$^{CD2}$ mice was not restored in IKKΔT$^{CD2}$ RIPK1$^{D138N}$ mice. These results suggest that γδ T cells do require IKK activity to repress RIPK1-dependent cell death pathways, but that blocking this pathway alone is insufficient to restore peripheral γδ T cell compartment, and IKK is required to mediate additional functions to support peripheral γδ T cells. They also provided evidence that γδ T cells are susceptible to necroptosis, and that the failure of CASPASE8 ablation to rescue IKK-deficient cells from cell death was due to a switch from apoptotic to necroptotic death in the absence of IKK expression.

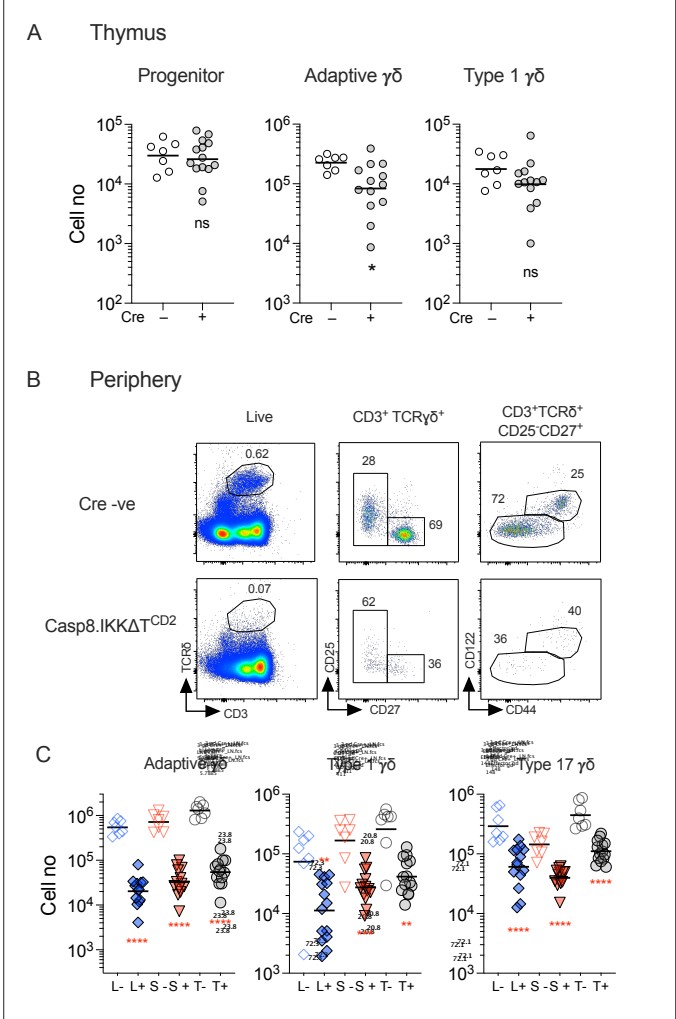

**Figure 2.** CASPASE8 ablation does not rescue peripheral γδ T cell compartment of IKKΔT[CD2] mice. Thymi, lymph nodes and spleen from Casp8.IKKΔT[CD2] mice (*n* = 13) and Cre –ve littermates (*n* = 7) were enumerated and analysed by flow. (**A**) Scatter plots are of total cell numbers of the indicated subset recovered from thymi of Casp8. IKKΔT[CD2] mice or Cre –ve littermates. (**B**) Representative flow plots from lymph nodes of the indicated mouse strains (rows), with the indicated electronic gates (columns). (**C**) Scatter plots are of total cell numbers recovered from lymph nodes (L), spleen (S), and total combined spleen and lymph nodes (T) of the indicated subset from Cre +ve (+) and Cre –ve (–) littermates. Horizontal bars are mean. Fractional representation of different γδ subsets in different organs is shown in *Figure 1—figure supplement 2B*. Data are pooled from multiple batches of mice analysed. Horizontal lines indicate mean. **p < 0.01, ****p < 0.0001, Mann–Whitney test.

The online version of this article includes the following source data for figure 2:

**Source data 1.** Cell enumerations of different gamma-delta subsets in different lymphoid organs.

## Adaptive and type 1 γδ T cells are highly susceptible to necroptosis

In order to directly test whether γδ T cells are susceptible to necroptosis, we next analysed mice in which *Casp8* alone was deleted in T cells, in Casp8ΔT[CD2] mice. Analysing γδ T cell development in the thymus revealed normal representation (*Figure 4A*) and numbers (*Figure 4B*) of progenitor and newly generated adaptive and type 1 γδ T cells. In the periphery, however, numbers of both adaptive and type 1 γδ T cells were profoundly reduced with only a small population of CD27[+] γδ T cells remaining in LNs and spleen (*Figure 4C, D*). In contrast, no significant difference in the total number of type 17 γδ T cells was apparent in the periphery, suggesting that CASPASE8 is not required for either development or maintenance of this subset (*Figure 4D*).

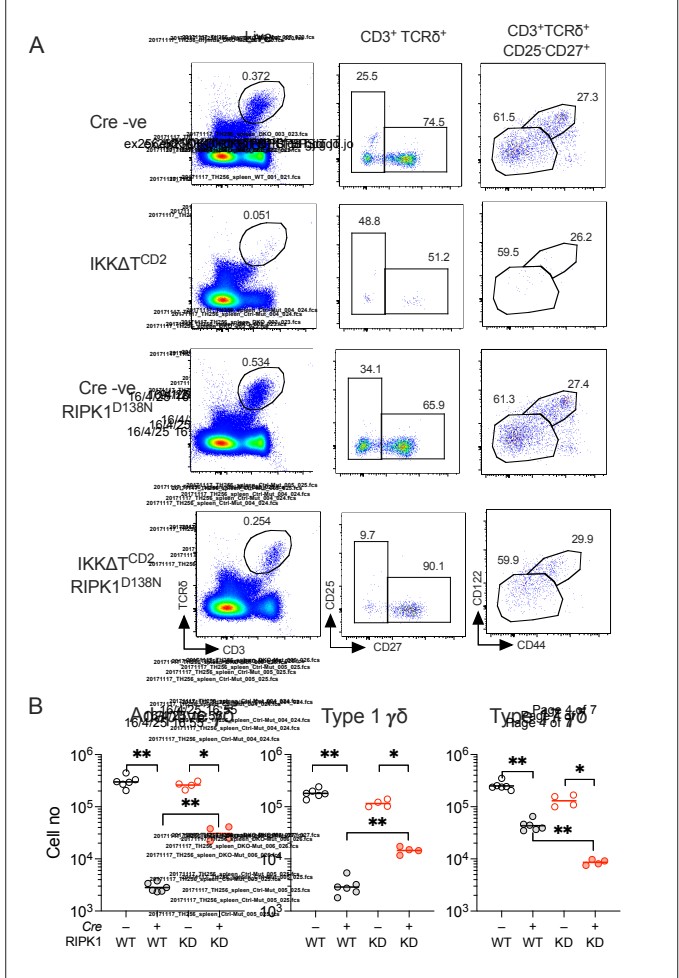

**Figure 3.** Kinase-dead RIPK1 mediates partial rescue of peripheral γδ T cell compartments of IKKΔT$^{CD2}$ mice. Lymph nodes and spleen from IKKΔT$^{CD2}$ ($n = 6$) IKKΔT$^{CD2}$RIPK1$^{D138N}$ mice ($n = 4$) and Cre –ve littermates ($n = 6$ and 4, respectively) were enumerated and analysed by flow. (**A**) Representative flow plots are of splenic cells from different strains (rows) with the indicated gates (columns). (**B**) Scatter plots are of total cell numbers of the indicated subset recovered from lymph nodes and spleen combined of the indicated IKKΔT$^{CD2}$ strain expressing either WT or kinase-dead (KD) RIPK1$^{D138N}$. Data are pooled from two independent experiments. Horizontal lines indicate mean.*$p < 0.05$, **$p < 0.01$, Mann–Whitney test.

The online version of this article includes the following source data for figure 3:

**Source data 1.** Cell enumerations of different gamma-delta subsets in different lymphoid organs.

---

To confirm that the loss of peripheral γδ T cells was due to necroptosis in the absence of *Casp8* expression, we generated Casp8ΔT$^{CD2}$ RIPK1$^{D138N}$ mice expressing kinase-dead RIPK1, to see if this would rescue cells from death. Analysing the periphery of this strain revealed normal representation (*Figure 5A*) and numbers (*Figure 5B*) of adaptive, type 1, and type 17 γδ T cells, confirming that loss of adaptive and type 1 subsets in Casp8ΔT$^{CD2}$ mice was the result of necroptosis in the absence of *Casp8* expression. We further confirmed that rescued cells in Casp8ΔT$^{CD2}$ RIPK1$^{D138N}$ mice were the functional counterparts of the same subsets in Cre –ve controls by assessing cytokine production in vitro. IL-17A and IFN-gamma production was similar in extent and restricted to the corresponding type 17 and type 1 subsets (*Figure 5C*).

## Alternative NF-κB activation is redundant for normal γδ T cell homeostasis

Our data demonstrated that generation of type 1 and long-term maintenance of adaptive γδ T cells was highly dependent upon IKK signalling. IKK kinase activity is required by T cells to both trigger

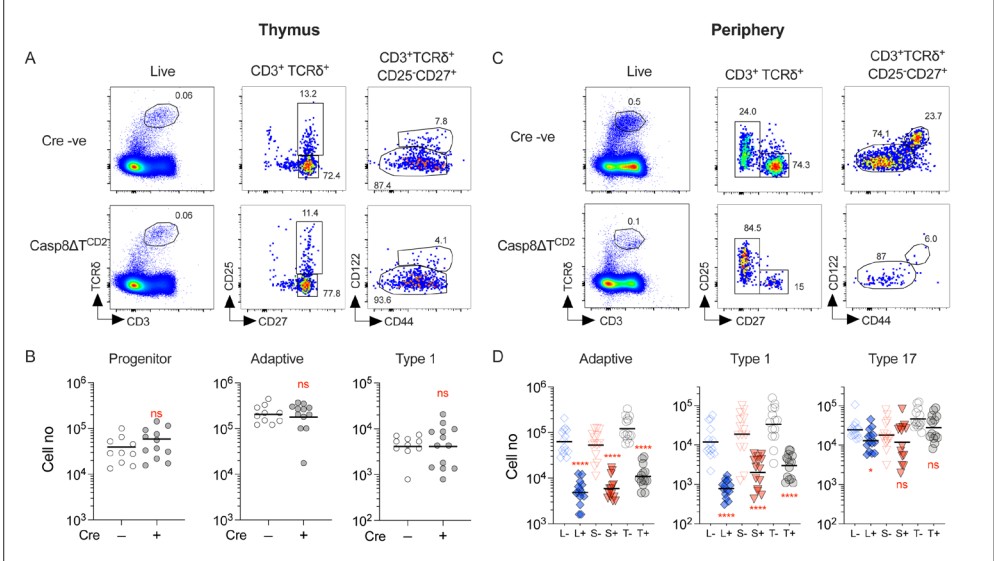

**Figure 4.** CASPASE8 expression is critical for long-term survival of peripheral γδ T cells. Thymi, lymph nodes, and spleen from Casp8ΔT^CD2 mice (*n* = 13) and Cre –ve littermates (*n* = 10) were enumerated and analysed by flow. (**A**) Representative flow plots are of thymocytes with the indicated gates, showing gates used to identify progenitor, adaptive, and type 1 subsets. (**B**) Scatter plots are of total cell numbers of the indicated subset from thymus of Casp8ΔT^CD2 mice (+) and Cre –ve littermates (–). (**C**) Representative flow plots illustrate gating strategy to identify adaptive, type 1, and type 17 subsets in lymph nodes (shown) and spleen from the indicated strains. (**D**) Scatter plots are of total cell numbers recovered from lymph nodes (L), spleen (S), and total combined spleen and lymph nodes (T), of the indicated subsets from Cre +ve (+) and Cre –ve (–) littermates. Horizontal bars are means. Fractional representation of different γδ subsets in different organs is shown in *Figure 1—figure supplement 2C*. Data are pooled from five independent experiments. Horizontal lines indicate mean. n.s. – not significant, ****p < 0.0001, Mann–Whitney test.

The online version of this article includes the following source data for figure 4:

**Source data 1.** Cell enumerations of different gamma-delta subsets in different lymphoid organs.

NF-κB activation and to directly block extrinsic cell death pathways by inhibiting RIPK1 activity (*Blanchett et al., 2021*). Kinase-dead RIPK1 only achieved a modest rescue of peripheral γδ T cell numbers, suggesting that these cells also require the IκB kinase activity of IKK for their persistence, and that they also require NF-κB. We therefore dissected the specific role of NF-κB for γδ T cell homeostasis. IKK proteins regulate activation of both canonical NF-κB, mediated by Rela/p50 and cRel/p50 heterodimers, and non-canonical or alternative NF-κB, mediated by RelB/p52 dimers. A trimeric complex of IKK1, IKK2 and regulatory subunit NEMO is responsible for triggering canonical activation, while alternative NF-κB is triggered exclusively by homodimers of IKK1. In IKKΔT^CD2 mice, γδ T cells are unable to activate either of these NF-κB pathways.

To determine if alternative NF-κB pathways contribute to the phenotype of IKKΔT^CD2 mice, we analysed IKK1ΔT^CD2 mice that lack IKK1 expression while retaining IKK2 expression. Canonical NF-κB activation in T cells is largely normal in the absence of IKK1, while alternative NF-κB is completely blocked (*Lawrence, 2009*). Analysing thymus from these mice showed that development of both adaptive and type 1 γδ T cells was normal (*Figure 6A, B*). Similarly, numbers and representation of adaptive, type 1, and type 17 γδ T cells in the periphery were also normal in these mice as compared with Cre –ve littermates. Therefore, this suggests that alternative NF-κB pathways are redundant for both the development and persistence of γδ T cells.

## Canonical NF-κB is essential for maintenance of peripheral γδ T cells

The phenotype of IKK1ΔT^CD2 mice appeared to exclude a role for alternative NF-κB pathways in regulating γδ T cell homeostasis. Therefore, we next sought to dissect the role of canonical NF-κB pathways. The p50 subunit that derives from processing of NFKB1 lacks a transactivation domain, so while it can bind DNA, it cannot activate transcription alone. Therefore, combined ablation of cREL and RELA

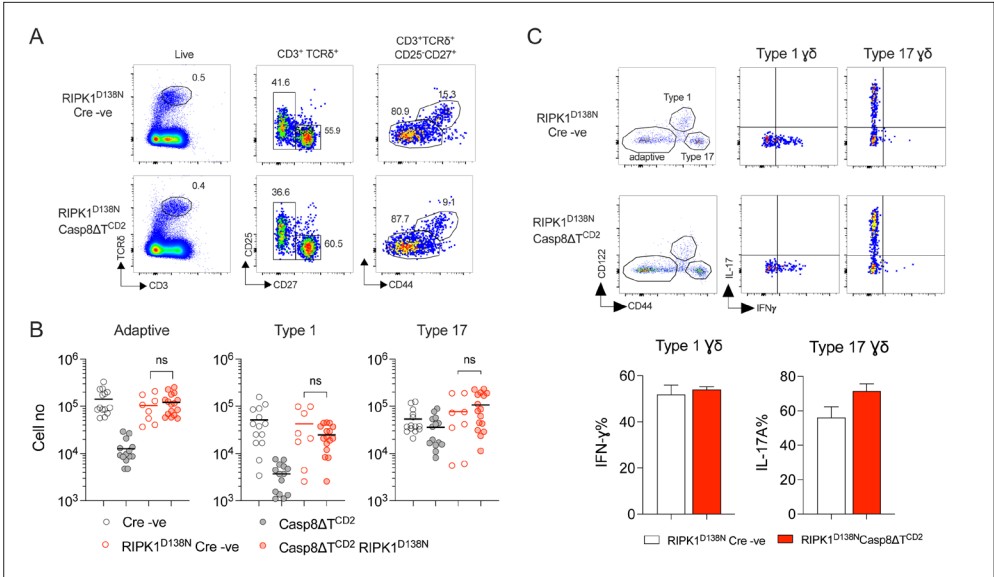

**Figure 5.** Kinase-dead RIPK1 fully restores the peripheral γδ T cell compartments of Casp8ΔT^CD2 mice. Lymph nodes and spleen from Casp8ΔT^CD2 RIPK1^D138N mice (*n* = 16) and Cre –ve RIPK1^D138N littermates (*n* = 8) were enumerated and analysed by flow. (**A**) Representative flow plots are of lymph nodes from Casp8ΔT^CD2 RIPK1^D138N mice and Cre –ve RIPK1^D138N littermates with the indicated gates applied (columns). (**B**) Scatter plots are of total cell numbers of the indicated subsets from both lymph nodes and spleen combined from Casp8ΔT^CD2 RIPK1^D138N mice and Cre –ve RIPK1^D138N littermates (red symbols) and cell numbers from Casp8ΔT^CD2 RIPK1WT strains described in *Figure 4*, for direct comparison. (**C**) Lymph node cells were stimulated in vitro with calcium ionophore and phorbyl esters for 4 hr with brefeldin A, and then analysed for the indicated intracellular cytokines. Representative flow plots illustrate gating strategy to identify adaptive, type 1, and type 17 subsets on the basis of CD44 and CD122 expression. Quad gates for cytokine detection were set against negative controls of matched unstimulated cells. Bar charts are of total % of cells stained for IFN-gamma or IL-17A. Data are pooled from multiple batches of mice analysed (**A, B**) or are pooled from four independent experiments (**C**). n.s. – not significant, Mann–Whitney test.

The online version of this article includes the following source data for figure 5:

**Source data 1.** Cell enumerations of different gamma-delta subsets and representation of cytokine producing cells.

is sufficient to completely block canonical NF-κB pathways. Amongst conventional αβ T cells, cREL expression is redundant for naive T cells that can be supported by RELA alone, while memory pheno-type T cells depend upon both cREL and RELA expression for their generation and/or persistence (*Webb et al., 2019*; *Zheng et al., 2003*). Therefore, we analysed mice whose T cells lacked expression of only RELA (RelaΔT^CD2), mice with germline *Nfkb1* deficiency (*Nfkb1*^−/−), combined T cell deficiency of RELA and germline *Nfkb1* deficiency (RelaΔT^CD2*Nfkb1*^−/−) and combined T cell-specific ablation of both RELA and cREL (Rela.RelΔT^CD2). We first analysed the thymus of these strains to assess the devel-opment of adaptive and type 1 γδ T cells. Numbers of progenitor and adaptive γδ T cells appeared largely normal (*Figure 7A*), while there was some statistical evidence of a modest (~2-fold) reduction in absolute numbers of both these subsets in some of the REL-deficient strains. This was in contrast to the absence of a similar phenotype in IKKΔT^CD2 mice (*Figure 1*). We therefore also compared repre-sentation of progenitors γδ T cells with DN3 cell numbers, which reflect upstream progenitor pool. This ratio was similar across strains (*Figure 7A*), suggesting that the observed modest reductions were not cell intrinsic but may rather reflect differences in overall thymic cellularity from these strains. In contrast, analysing representation and numbers of type 1 γδ T cells revealed similar reductions as observed in IKK1ΔT^CD2 mice, suggesting that development of this subset is indeed dependent upon canonical NF-κB. Furthermore, analysing the impact of different REL subunit ablations suggested that development of this subset was highly dependent upon NF-κB, since even ablation of RELA alone was sufficient to reduce the numbers of newly generated cells in the thymus, while combined ablation of both RELA and cREL resulted in the greatest reduction in numbers of type 1 γδ T cells (*Figure 7A*).

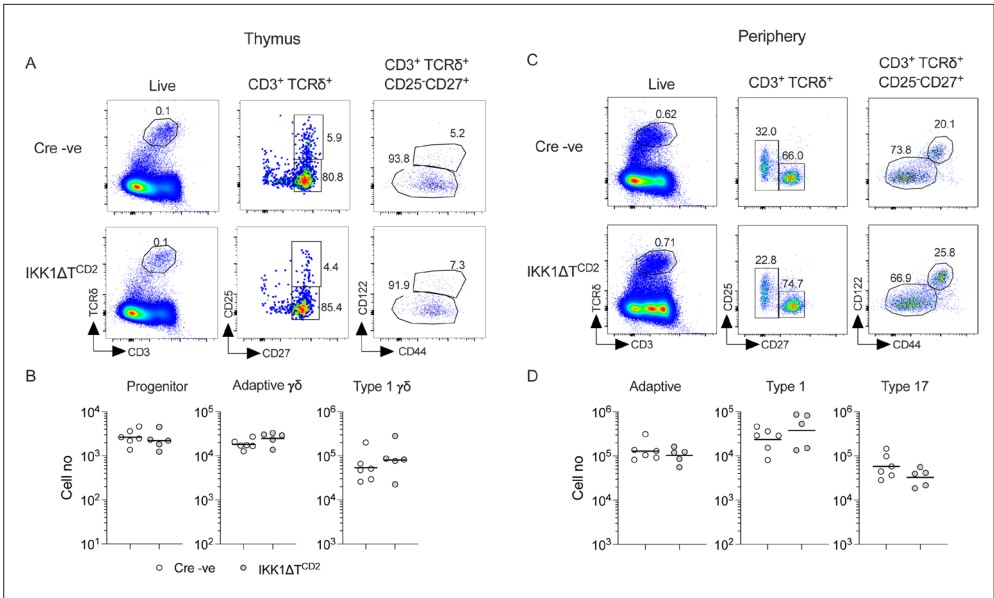

**Figure 6.** Alternative NF-κB signalling is redundant for generation and maintenance of γδ T cell compartments. Thymi, lymph nodes, and spleen from IKK1ΔT^CD2 mice (*n* = 5) and Cre –ve littermates (*n* = 6) were enumerated and analysed by flow. (**A**) Representative flow plots are of thymocytes from IKK1ΔT^CD2 mice and Cre –ve littermate, showing gates used to identify progenitor, adaptive, and type 1 subsets intrathymically. (**B**) Scatter plots are of total cell numbers of the indicated subset from thymus. (**C**) Representative flow plots illustrate gating strategy to identify adaptive, type 1, and type 17 subsets in lymph nodes (shown) and spleen. (**D**) Scatter plots are of total cell numbers recovered from both spleen and lymph nodes of the indicated subset. Data are pooled from two independent experiments.

The online version of this article includes the following source data for figure 6:

**Source data 1.** Cell enumerations of different gamma-delta subsets in different lymphoid organs.

---

Finally, we analysed the peripheral compartments of different REL-deficient strains to determine the specific requirements for NF-κB by different subsets. In mice lacking RELA alone, RELA/NFKB1, or RELA/cREL, both type 1 and type 17 γδ T cells were substantially reduced (*Figure 7B*). The reduction in numbers of type 1 γδ T cells in the periphery of these mice mirrored reductions observed in the thymus, further reinforcing the view that NF-κB signalling was essential for their normal thymic generation. The reduction of type 17 γδ T cells suggests that NF-κB is also necessary for the development and/or maintenance of this population. Type 1 and type 17 γδ T cells were similarly reduced in all three REL-deficient strains, suggesting a strong dependence upon NF-κB. Numbers of peripheral adaptive γδ T cells were also profoundly reduced in RELA/cREL-deficient mice, but there was evidence for redundancy between subunits, since specific ablation of RELA alone was well tolerated (*Figure 7B*). Overall, these data show that maintenance of replete peripheral γδ T cell compartments is highly dependent on tonic NF-κB signals either for development or maintenance of different subsets.

## Discussion

The IKK complex is a critical regulator of cell death, differentiation, and inflammation in multiple cell types, including αβ T cells. In the present study, we sought to understand whether IKK signalling was also important for the development and maintenance of γδ T cells, which are implicated as early responders in a host of different inflammatory settings. We found that IKK was critical for establishing a mature γδ T cell compartment, but found evidence for contrasting roles and mechanisms of downstream pathways for the development and maintenance of different subsets.

The inhibitor of κB kinase activity of the IKK complex is critical for activating NF-κB signalling, so we were able to demonstrate the essential role of this transcription factor by two independent means – either ablating the upstream IKK complex or direct ablation of canonical REL subunits. Numbers of type 1 γδ T cells in the thymus were substantially reduced in both settings, suggesting that generation

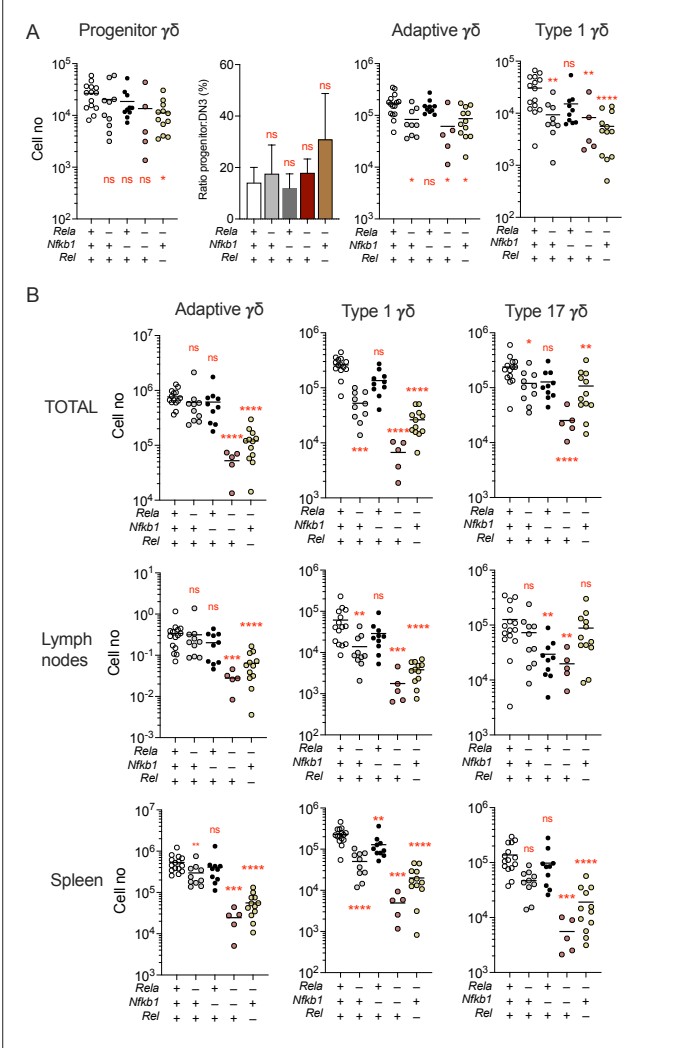

**Figure 7.** Canonical NF-κB signalling is essential for development of type 1 and maintenance of adaptive γδ T cell compartments. Thymi, lymph nodes, and spleen from RelaΔT^CD2 (n = 10), Nfkb1^−/− (n = 10), RelaΔT^CD2 Nfkb1^−/− mice (n = 5), Rela.RelΔT^CD2 (n = 12), and Cre –ve littermates (n = 14) were enumerated and analysed by flow. (**A**) Scatter plots are of total cell numbers of the indicated subset from thymus, while bar charts show the ratio of progenitor:DN3 subsets for the indicated strains.+ indicates WT allele, – indicates gene deletion for the different REL subunits. (**B**) Scatter plots are of total cell numbers recovered from both spleen and lymph nodes combined (top row), lymph nodes (middle row), or spleen (bottom row) of the indicated subset from mice with different REL subunit deletions (+ indicates WT, – indicates deletion). Fractional representation of different γδ subsets in different organs is shown in **Figure 1—figure supplement 2D**. Data are pooled from multiple batches of mice analysed. n.s. – not significant, *p < 0.05, **p < 0.01, ****p < 0.0001 by Mann–Whitney test.

The online version of this article includes the following source data for figure 7:

**Source data 1.** Cell enumerations of different gamma-delta subsets in different lymphoid organs.

---

of this pre-differentiated subset requires NF-κB signalling to support their ontogeny. In contrast, NF-κB signalling was redundant for both specification of progenitors to a γδ T cell lineage and generation of adaptive γδ T cells, as numbers of progenitors and adaptive γδ T cells were largely normal in both REL- and IKK-deficient strains. Strikingly, however, mature adaptive γδ T cells were largely absent in both REL- and IKK-deficient mice, revealing that NF-κB is critical for maintenance of mature γδ T cells. Assessing the role of different REL subunits also revealed distinct requirements for development vs maintenance that mirrored preferences observed in αβ T cells. Generation of effector αβ T cells is highly dependent upon NF-κB, since loss of cREL alone is sufficient to block development of αβ T cell memory (**Webb et al., 2019**; **Zheng et al., 2003**). Similarly, we found that loss of RELA alone was

sufficient to impair development of type 1 γδ T cells, suggesting the cREL and p50 alone are insufficient for their development. Furthermore, in naive αβ T cell, there is evidence of redundancy between REL subunits. Expression of cREL and p50 in the absence of RELA is sufficient to maintain naive αβ T cells, while expression of cREL alone, in RELA/p50 double knockouts, is not (*Webb et al., 2019*). We also found this to be the exact case for adaptive γδ T cells. These subtly distinct requirements for NF-κB probably reflect both distinct biological processes involved, ontogeny vs maintenance, but also potentially the receptors involved in triggering signalling. TNFRSF members appear to be implicated in triggering survival signalling in naive αβ T cells (*Silva et al., 2014*) while generation of differentiated αβ T effectors is dependent on strong agonist triggers from TCR and also TNFRSF members (*Layzell et al., 2025*). This is also true of Foxp3⁺ regulatory T cells, that require both RELA and cREL for their development (*Isomura et al., 2009*; *Messina et al., 2016*), and are induced by a combination of both agonist TCR and TNFRSF signalling, amongst other receptors.

In αβ T cells, IKK also plays a crucial role in controlling cell survival directly by its repressive activity upon the cell death regulator, RIPK1. Developing thymocytes do not require NF-κB signalling for development but do become exquisitely sensitive to TNF-induced cell death in the absence of IKK (*Webb et al., 2019*), while mature naive T cells require IKK expression to both repress RIPK1 and activate an NF-κB dependent survival signal. The balance between control of NF-κB and blocking cell death by IKK appears different in γδ T cells, as does the control of cell death processes themselves. Neither αβ nor γδ thymocytes undergo necroptosis in the absence of CASPASE8, while, in stark contrast to resting αβ T cells, adaptive, and type 1 γδ T cells in the periphery were exquisitely sensitive to the induction of necroptosis in the absence of CASPASE8. This appears to be determined in part by expression of MLKL that is present in γδ T cells but absent from resting αβ T cells (ImmGen browser enquiry; *Heng et al., 2008*). Using kinase-dead RIPK1, it was possible to test the importance of regulation of RIPK1-dependent cell death pathways by IKK, since this RIPK1 mutant blocks both necroptosis and apoptosis, while CASPASE8 ablation served to block apoptosis but also redirected cell death processes to a necroptotic modality. IKKΔTᶜᴰ² mice expressing kinase-dead RIPK1 did exhibit some significant, albeit incomplete, rescue of the peripheral T cell compartment. The phenotype of REL-ablated mice confirmed that IKK is required to trigger an NF-κB dependent survival programme. The relative importance of IKK-induced NF-κB vs IKK-mediated repression of RIPK1 for the maintenance of peripheral γδ T cells is not easy to disentangle. The extent of peripheral γδ T loss in the absence of IKK was substantially greater than observed in REL-deficient strains. Kinase-dead RIPK1 restored peripheral γδ T cell levels to similar to those observed in REL-deficient mice, suggesting that the impact of IKK deficiency is indeed a compound effect of losing both IKK-induced NF-κB activity and loss of IKK-dependent repression of RIPK1. It would be interesting to test the impact of losing IKK-dependent control of RIPK1 in the context of normal NF-κB activation upon peripheral γδ T cell compartment.

In order to target gene deletion prior to γδ T cell commitment, we employed the use of huCD2ⁱᶜʳᵉ that is expressed from common lymphoid progenitors. This will inevitably also target deletion in αβ T cells, the consequences of which have already been extensively characterised for IKK, REL and CASPASE8 genes (*Islam et al., 2025*; *Webb et al., 2019*; *Webb et al., 2016*). This does raise the question of whether alterations to the αβ T cell compartment affect the phenotypes we describe amongst γδ T cells. Studies of TCRα knockout mice confirm that development and maintenance of γδ T cells does not depend upon the presence of αβ T cells (*Hayday and Tigelaar, 2003*; *Philpott et al., 1992*). However, in the absence of peripheral αβ T cells, as is the case in IKKΔTᶜᴰ² mice, there will be reduced competition for cytokines that both lineages rely upon, such as IL-7 and IL-15 (*French et al., 2005*). Potential alterations to homeostatic niches did not appear to impact our results, however. The loss of γδ T cells in the strains we investigated here occurred both in the presence and absence of αβ T cells. We observed normal thymic development of γδ T cells but their complete absence from peripheral compartments in both IKKΔTᶜᴰ² and Casp8.IKKΔTᶜᴰ² strains, in spite of the fact that development and peripheral population of αβ T cell compartment is completely blocked in IKKΔTᶜᴰ² mice but restored in Casp8.IKKΔTᶜᴰ² mice. Therefore, an absence of competition for IL-7 and IL-15 in IKKΔTᶜᴰ² mice is not sufficient to overcome survival defects of γδ T cells we observed in these strains and the defects in the γδ T cell compartment do appear cell intrinsic.

Type 17 γδ T cells are primarily generated during embryonic stages of development (*Muñoz-Ruiz et al., 2017*), and so we could not directly observe their intrathymic generation in adult mice. As such,

the presence or absence of cells in adult mice could reflect either developmental processes and/or their defective survival. Nevertheless, there were still some useful conclusions we could draw. The maintenance of the mature type 17 γδ T cells does depend on NF-κB signalling. Peripheral numbers of type 17 γδ T cells were reduced by around two-thirds in both IKK- and REL-deficient strains. However, in contrast to other subsets, there was no evidence that kinase-dead RIPK1 mediated any rescue of type 17 subset, and they did not appear to be sensitive to necroptosis in the absence of CASPASE8. Their numbers were also not restored in Casp8.IKKΔT$^{CD2}$ mice, strongly suggesting that the sole defect in IKK-deficient strains was the absence of NF-κB activation. An important caveat here is that REL and IKK-deficient strains also lack effector αβ T cell populations in the absence of NF-κB activation (*Webb et al., 2019*). There are reports of 'trans-conditioning', whereby development of type 17 γδ T cells is promoted by type 17 αβ T cells (*Do et al., 2012*). Therefore, it is possible that a failure of trans-conditioning by αβ type 17 T cells is a contributing factor in REL- and IKK-deficient strains. However, our findings are also consistent with other studies that show that RelA is involved in the development of type 17 γδ T cells (*Powolny-Budnicka et al., 2011*). The same study also implicated RelB transcription factor in the development of type 17 γδ T cells. This appears at odds with our findings of normal development of these cells in IKK1-deficient mice, since IKK1 homodimers mediate activation of the alternative NF-κB pathway. It is possible that type 17 γδ T cells are still RelB dependent in IKK1ΔT$^{CD2}$ mice but that there is redundancy between IKK subunits to activate RelB containing NF-κB dimers, permitting development of these cells. The stark contrast in sensitivity of type 17 and other subsets to necroptosis in the absence of CASPASE8 was surprising, as they all appear to express MLKL (ImmGen browser enquiry; *Heng et al., 2008*). It is possible this may reflect altered cell death control by the distinct fetal liver-derived progenitors from which type 17 cells are derived. Whichever the case, if confirmed, it would be of interest to identify the mechanism by which these cells are resistant to necroptosis, since they appear to express the basic elements required to form the necrosome.

In conclusion, our study reveals the complexity of regulation of inflammatory and cell death pathways in T cells of the adaptive immune system. In other cell types and tissues, IKK mediates well-defined pro-inflammatory and pro-survival functions. In contrast, in T cells, the functions of IKK and how cell death pathways are regulated vary with differentiation state and lineage. IKK-dependent repression of RIPK1 is critical for thymic development of αβ T cells, but is redundant for thymic development in γδ T cells. In common with αβ T cells, mature γδ T cells require IKK to both repress RIPK1-dependent cell death and activate NF-κB survival, with perhaps a more dominant requirement for the latter function. We found evidence that intrathymic development of type 1 γδ T cells did depend upon NF-κB. Their development is thought to be driven by strong 'agonistic' TCR signalling, and it is perhaps significant that other agonist-selected populations, such as NK T cells and Foxp3$^+$ regulatory T cells, are also dependent upon NF-κB signalling (*Isomura et al., 2009*; *Sivakumar et al., 2003*), implying a conserved developmental mechanism governs generation of such agonist-selected populations in the thymus. It is also interesting to note that, like γδ T cells, NK T cells, and Treg are also susceptible to necroptosis in the absence of CASPASE8 expression (*Islam et al., 2025*; *Teh et al., 2022*), in contrast to resting αβ T cells, which are largely resistant. This does contrast with adaptive γδ T cells that do become exquisitely sensitive to necroptosis induction as soon as they enter the periphery. This difference may reflect the role of γδ T cells as first line early responders in immune responses, that may expose them to microbial interference of cell death pathways, that necroptosis has evolved to resist. Whichever is the case, a detailed understanding of these pathways will be critical to understand the full impact of therapies that target these potent inflammatory pathways, and that will have diverse and complex impact on adaptive immunity.

## Materials and methods
### Mice
Mice with the following mutations were used in this study; *B6.129-Casp8$^{tm1Hed}$/J* (*Casp8$^{fx}$*), *Ikbkbtm2Mka* (*Li et al., 2003*) (*Ikbkb$^{fx}$*), *Chukrm1Mpa* (*Gareus et al., 2007*) (*Chuk$^{fx}$*), *Relatm1Asba* (*Steinbrecher et al., 2008*) (*Rela$^{fx}$*), *B6.129S1-Reltm1Ukl/J* (*Heise et al., 2014*) (*Rel$^{fx}$*), *Nfkb1tm1Bal* (*Nfkb1$^{-/-}$*), Cre transgenes expressed under the control of the human CD2, *B6.Cg-Tg(CD2-icre)4Kio/J* (huCD2$^{iCre}$) (*de Boer et al., 2003*) mice with a D138N mutation in *Ripk1*, *B6.129-Ripk1$^{tm1Geno}$/J* (RIPK1$^{D138N}$) (*Newton et al., 2014*). The following strains were bred using these alleles for this study; *Chuk$^{fx/}$*

<sup>fx</sup> *Ikbkb*<sup>fx/fx</sup> huCD2<sup>iCre</sup> (IKKΔT<sup>CD2</sup>), *Chuk*<sup>fx/fx</sup> huCD2<sup>iCre</sup> (IKK1ΔT<sup>CD2</sup>), *Chuk*<sup>fx/fx</sup> *Ikbkb*<sup>fx/fx</sup> *Casp8*<sup>fx/fx</sup> huCD2<sup>iCre</sup> (Casp8.IKKΔT<sup>CD2</sup>), *Chuk*<sup>fx/fx</sup> *Ikbkb*<sup>fx/fx</sup> huCD2<sup>iCre</sup> *Ripk1*<sup>D138N</sup> (IKKΔT<sup>CD2</sup> RIPK1<sup>D138N</sup>), *Casp8*<sup>fx/fx</sup> huCD2<sup>iCre</sup> (Casp8ΔT<sup>CD2</sup>), *Casp8*<sup>fx/fx</sup> huCD2<sup>iCre</sup>*Ripk1*<sup>D138N</sup> (Casp8ΔT<sup>CD2</sup>RIPK1<sup>D138N</sup>), *Rela*<sup>fx</sup> huCD2<sup>iCre</sup> (RelaΔT<sup>CD2</sup>), *Rela*<sup>fx/fx</sup> *Rel*<sup>fx/fx</sup>huCD2<sup>iCre</sup> (Rela.RelΔT<sup>CD2</sup>), and *Rela*<sup>fx/fx</sup> huCD2<sup>iCre</sup>*Nfkb1*<sup>−/−</sup>(RelaΔT<sup>CD2</sup> *Nfkb1*<sup>−/−</sup>). All mice were bred in the Comparative Biology Unit of the Royal Free UCL campus and at Charles River laboratories, Manston, UK. Animal experiments were performed according to the institutional guidelines and Home Office regulations under project licence PP2330953.

## Flow cytometry and electronic gating strategies

Flow cytometric analysis was performed with $10 \times 10^6$ thymocytes, $5 \times 10^6$ LN or spleen cells. Cell concentrations of thymocytes, LN, and spleen cells were determined with a Scharf Instruments Casy Counter. Cells were incubated with saturating concentrations of antibodies in 100 µl of Dulbecco's phosphate-buffered saline (PBS) containing bovine serum albumin (BSA, 0.1%) for 1 hr at 4°C followed by two washes in PBS-BSA. Panels used the following mAb: BV421-conjugated antibody against CD27 (Biolegend) (RRID:AB_11150782), PE-conjugated antibody against CD127 (Thermo Fisher Scientific) (RRID:AB_1659672), BV785-conjugated CD44 antibody (Biolegend) (RRID:AB_2566588), BUV395-conjugated antibody against CD25 (BD horizon) (RRID:AB_2868827), PE-Cy7-conjugated antibody against CD122 (Biolegend) (RRID:AB_2563460), PerCP-cy5.5-conjugated antibody against CD3 (eBioscience) (RRID:AB_2572434), APC-conjugated antibody against TCRδ (eBioscience) (RRID:AB_469968), FITC-conjugated antibody against IFN-γ (BD Biosciences) (RRID:AB_395366), PE-Cy7-conjugated antibody against IL-17A (eBioscience) (RRID:AB_2573694). Cell viability was determined using LIVE/DEAD cell stain kit (Invitrogen Molecular Probes), following the manufacturer's protocol. Multi-colour flow cytometric staining was analysed on a LSRFortessa (Becton Dickinson) instrument, and data analysis and colour compensations were performed with FlowJo V10 software (TreeStar). The following gating strategies were used: γδ progenitor T cells – CD3<sup>+</sup> TCRδ<sup>hi</sup> CD25<sup>+</sup> CD27<sup>+</sup>, adaptive γδ T cells – CD3<sup>+</sup>TCRδ<sup>+</sup>CD25<sup>−</sup>CD27<sup>+</sup>CD44<sup>lo</sup>CD122<sup>lo</sup>, type 1 γδ T cells – CD3<sup>+</sup>TCRδ<sup>+</sup>CD25<sup>−</sup>CD27<sup>+</sup>CD44<sup>+</sup>CD122<sup>+</sup>, and type 17 γδ T cells – CD3<sup>+</sup>TCRγ<sub>δ</sub><sup>+</sup>CD25<sup>−</sup>CD27<sup>−</sup>, respectively.

## Intracellular cytokine staining

LN T cells (cervical, auxiliary, brachial, inguinal, and mesenteric) were cultured at 37°C with 5% $CO_2$ in RPMI-1640 (Gibco, Invitrogen Corporation, CA) supplemented with 10% (vol/vol) fetal bovine serum (Gibco Invitrogen), 0.1% (vol/vol) 2-mercaptoethanol βME (Sigma-Aldrich), and 1% (vol/vol) penicillin–streptomycin (Gibco Invitrogen) (RPMI-10). Approximately $5 \times 10^6$ LN cells were stimulated with a protein transport inhibitor cocktail (Brefeldin and Monensin) at 12.6 µM concentration and cell stimulation cocktail (PMA and Ionomycin) at 1.42 µM concentration resuspended in RPMI. The stimulated cells were incubated for 4 hr, followed by the cell surface and intracellular cytokine staining.

## Statistics

Statistical analysis and bar charts were performed using GraphPad Prism 9.2. Column data compared by two-tailed *t*-test (non-parametric) Mann–Whitney *t*-test. Results were denoted as ns = nonsignificant, *$p < 0.05$, **$p < 0.01$, ***$p < 0.001$, and ****$p < 0.0001$.

## Acknowledgements

We thank UCL Comparative Biology Unit staff for assistance with mouse breeding and maintenance. We thank the following for generously sharing their mouse strains: Prof Manolis Pasparakis for Chuk conditional strain, Prof Michael Karin for Ikbkb conditional strain, Prof Vishva Dixit for the RIPK1<sup>D138N</sup> strain, and Prof Albert Baldwin for Rela conditional strain. The authors declare no competing financial interests. The work in the Seddon lab is supported by the Medical Research Council UK under programme code MR/P011225/1. FIS was supported by a scholarship from the Commonwealth Scholarship Commission of the United Kingdom.

# Additional information

## Funding

| Funder | Grant reference number | Author |
|---|---|---|
| Medical Research Council | MR/P011225/1 | Cayman Williams<br>Louise V Webb<br>Ines Boal-Carvalho<br>Benedict Seddon |
| Commonwealth Scholarship Commission | | Farjana Islam |

The funders had no role in study design, data collection, and interpretation, or the decision to submit the work for publication.

## Author contributions

Farjana Islam, Conceptualization, Data curation, Formal analysis, Funding acquisition, Investigation, Methodology, Project administration, Visualization, Writing – review and editing; Cayman Williams, Investigation, Methodology, Data curation; Thea Hogan, Methodology, Data curation, Visualization; Louise V Webb, Methodology, Data curation; Ines Boal-Carvalho, Formal analysis, Investigation, Supervision, Data curation, Project administration, Writing – review and editing; Benedict Seddon, Conceptualization, Data curation, Formal analysis, Funding acquisition, Methodology, Project administration, Supervision, Visualization, Writing – original draft

## Author ORCIDs

Farjana Islam ⓘ https://orcid.org/0000-0001-8274-2021
Louise V Webb ⓘ https://orcid.org/0009-0000-8862-0199
Benedict Seddon ⓘ https://orcid.org/0000-0003-4352-3373

## Ethics

Animal experiments were performed according to institutional guidelines and Home Office regulations under project licence PP2330953.

Reviewer #1 (Public review): https://doi.org/10.7554/eLife.108940.3.sa1
Reviewer #2 (Public review): https://doi.org/10.7554/eLife.108940.3.sa2
Author response https://doi.org/10.7554/eLife.108940.3.sa3

# Additional files

## Supplementary files

MDAR checklist

## Data availability

Processed cell counts are provided as supplementary source data files. Enquiries regarding access for raw flow cytometry data should be directed to the corresponding author. We will honour requests that do not conflict with our ongoing as yet unpublished studies.

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
