## [Editor Report · eLife Assessment]

This study reports **important** findings regarding the role of the NF-kB signaling pathway in the development and long-term survival of gamma delta T cells. The authors report disparate roles of IKK-dependent NF-kB activation in the development and long-term survival of gamma delta T cell subsets. The approach and methodology employed is **convincing**. This work will be of great interest to immunologists interested in innate-like T cell biology and in T cell development.

---

## [Referee Report · Reviewer #1 (Public review)]

Summary:

The NF-kB signaling pathway plays a critical role in the development and survival of conventional alpha beta T cells. Gamma delta T cells are evolutionarily conserved T cells that occupy a unique niche in the host immune system and that develop and function in a manner distinct from conventional alpha beta T cells. Specifically, unlike the case for conventional alpha beta T cells, a large portion of gamma delta T cells acquire functionality during thymic development, after which they emigrate from the thymus and populate a variety of mucosal tissues. Exactly how gamma delta T cells are functionally programmed remains unclear. In this manuscript, Islam et al., use a wide variety of mouse genetic models to examine the influence of the NF-kB signaling pathway on gamma delta T cell development and survival. They find that the inhibitor of kappa B kinase complex (IKK) is critical to the development of gamma delta T1 subsets, but not adaptive/naïve gamma delta T cells. In contrast, IKK-dependent NF-kB activation is required for their long-term survival. They find that caspase 8-deficiency renders gamma delta T cells sensitive to RIPK1-mediated necroptosis and they conclude that IKK repression of RIPK1 is required for the long-term survival of gamma delta T1 and adaptive/naïve gamma delta T cells subsets. These data will be invaluable in comparing and contrasting the signaling pathways critical for the development/survival of both alpha beta and gamma delta T cells.

Comments on revisions:

The word adaptive is misspelt throughout most figures.

---

## [Referee Report · Reviewer #2 (Public review)]

This study presents a comprehensive genetic dissection of the role of IKK signaling in the development and maintenance of lymphoid gd T cells. By employing a variety of conditional and mutant mouse models, the authors demonstrate that IKK-dependent NF-κB activation is essential for the generation of type 1 gd T cells, while adaptive gd T cells require this pathway primarily for long-term survival. The use of multiple complementary genetic strategies, including IKK deletion and modulation of RIPK1 and CASPASE8 activity, provides robust mechanistic insight into subset-specific regulation of gd T cell homeostasis. Overall, the study provides mechanistic insight for IKK-dependent regulation of gd T cell development and peripheral maintenance.

Comments on revisions:

Thank you for your comments and clarifications.

---

## [Author Response]

The following is the authors’ response to the original reviews

**Public Reviews:**

**Reviewer #1 (Public review):**
(1) The authors appear to be excluding a significant fraction of the TCRlow gamma delta T cells from their analysis in Figure 1A. Since this population is generally enriched in CD25+ gamma delta T cells, this gating strategy could significantly impact their analysis due to the exclusion of progenitor gamma delta T cell populations.

We were cautious in our gating strategy since the TCR𝛿+ CD3e+ subset is rather small and so low signal/background noise ratio can be an issue if the gates used are too broad/generous. There is some inevitable low level background staining with the TCR𝛿 that sits just above the bulk of the negative population and is CD3ε -ve. Although this background represents a tiny fraction of total cells, we were wary of gate contamination into our TCR𝛿+ CD3e^+^ subset and we wanted a gating strategy that could be applied across other organs too. We do not, however, believe this conservative strategy is impacting on measurements progenitor numbers across strains or our conclusions, since the size of this progenitor population in the various IKKΔT^CD2^ and Casp8ΔT^CD2^ strains was never impacted by the mutations. But to reassure the reviewer, we show our conservative gate as compared with a very broad TCR𝛿 gate and see we are not missing a substantial population of CD25+ cells just below our gate. This also helps illustrate how close the background from the CD27^int^ expressing αβ thymocytes (right column) comes to the TCR𝛿+ CD3+ gate and the importance of tight lineage gating.

(2) The overall phenotype of the IKKDeltaTCd2 mice is not described in any great detail. For example, it is not clear if these mice possess altered thymocyte or peripheral T cell populations beyond that of gamma delta T cells.Given that gamma delta T cell development has been demonstrated to be influenced by gamma delta T cells (i.e, trans-conditioning), this information could have aided in the interpretation of the data.

Apologies for not being clearer on this point. We have studied conventional αβ T cell development in these strains in considerable detail, and these studies are published and discussed in some detail in the introduction in paragraph 3 on page 3-4 and in cited references Schmidt-Supprian et al 2004, SIlva et al 2014, Xing et al 2016, Webb et al 2019, Carty et al 2023. These detail how IKK expression is critical for thymic development of αβ T cells and their peripheral survival, and dissects the role of NF-κB activation and cell death regulation by IKK. However, we now add new discussion (page 11-12) that considers the potential impact of altered αβ T cell development in the strains used for this study.

We agree that trans-conditioning is also an important consideration, since CD4 TH17 T cells can enhance type 17 𝛾𝛿 T cell development (10.1038/icb.2011.50). This is of relevance to the limited conclusions we draw concerning type 17 𝛾𝛿 T cells. The REL and IKK deficient strains do lack effector populations, including type 17 αβ T cells, so it is possible that the absence of type 17 αβ T cells in these strains does contribute to the modest impact of IKK deletion in the type 17 𝛾𝛿 subset. We now highlight this information and discuss in the manuscript (page 11-12).

Related to this, it would have been helpful if the authors provided a comparison of the frequencies of each of the relevant subsets, in addition to the numbers.

We now provide both the absolute frequencies of different 𝛾𝛿 subsets and their relative frequencies to one another, as supplementary figure 2. We still believe assessing absolute numbers is the gold standard, since the differential impact of gene deletions on the αβ T cell compartments in different strains will effect whether or not αβ T cells are present, and therefore overall representation of 𝛾𝛿 T cells can vary considerably between strains. Hence, absolute numbers are more reliable measure of cell abundance.

(3) The manner in which the peripheral gamma delta T cell compartment was analyzed is somewhat unclear. The authors appear to have assessed both spleen and lymph node separately. The authors show representative data from only one of these organs (usually the lymph node) and show one analysis of peripheral gamma delta T cell numbers, where they appear to have summed up the individual spleen and lymph node gamma delta T cell counts. Since gamma deltaT17 and gamma deltaT1 are distributed somewhat differently in these compartments (lymph node is enriched in gamma deltaT17, while spleen is enriched in gamma deltaT1), combining these data does not seem warranted. The authors should have provided representative plots for both organs and calculated and analyzed the gamma delta T cell numbers for both organs separately in each of these analyses.

We did of course process and calculate numbers of different subsets in both lymph nodes and spleen. Where we saw loss of peripheral 𝛾𝛿 subsets, or rescue, this was reflected in seperate analysis of both organs and we did not see any organs specific effects in the mouse strains analysed. We therefore took the initial view that presenting aggregate data was most efficient and least repetitive representation of data. However, we very much recognise the reviewers concern, and interest to see these data, so have now included representative plots across both organs for figure 1D, and show cell numbers of lymph nodes and spleen separately, as well as together, for figures 1, 2, 4 and 7, and these plots reflect the differences observed when we combined data. We did not break down the data for all figures (e.g. figures 3 and 5) as it was more cumbersome for more complex multi-strain comparisons and so attempt to balance clarity and transparency against unnecessary repetitive data presentation.

(4) The authors make extensive use of surrogate markers in their analysis. While the markers that they choose are widely used, there is a possibility that the expression of some of these markers may be altered in some of their genetic mutants. This could skew their analysis and conclusions. A better approach would have been to employ either nuclear stains (Tbx21, RORgammaT) or intracellular cytokine staining to definitively identify functional gamma deltaT1 or gamma deltaT17 subsets.

We did share a similar concern, but think this is not an issue where subsets disappear and are almost completely absent, such as in IKK1/2 KO and Casp8 KO settings. Where we saw rescue with RIPK1^D138N^ in Casp8ΔT^CD2^ strains, we were keen to demonstrate that the populations we saw restored did exhibit their expected function, and so confirmed this in figure 5C by intracellular cytokine staining after a short 4h restimulation in vitro. This also served to validate our gating strategy, since what we designated as Type 1 cells - CD27+CD122+CD44^int^ cells were the only source of IFN-gamma, while CD27–CD44^hi^ CD122^lo^ cells were the only source of IL-17. Adaptive/ naive cells made neither cytokine. So while we did not include nuclear stains, we were satisfied that the cytokine assays validated the gating strategy.

(5) The analysis and conclusion of the data in Figure 3A is not convincing. Because the data are graphed on log scale, the magnitude of the rescue by kinase dead RIPK1 appears somewhat overstated. A rough calculation suggests that in type 1 game delta T cells, there is ~ 99% decrease in gamma delta T cells in the Cre+WT strain and a ~90% decrease in the Cre+KD+ strain. Similarly, it looks as if the numbers for adaptive gamma delta T cells are a 95% decrease and an 85% decrease, respectively. Comparing these data to the data in Figure 5, which clearly show that kinase dead RIPK1 can completely rescue the Caspase 8 phenotype, the conclusion that gamma delta T cells require IKK activity to repress RIPK1-dependent pathways does not appear to be well-supported. In fact, the data seem more in line with a conclusion that IKK has a significant impact on gamma delta T cell survival in the periphery that cannot be fully explained by invoking Caspase8-dependent apoptosis or necroptosis. Indeed, while the authors seem to ultimately come to this latter conclusion in the Discussion, they clearly state in the Abstract that "IKK repression of RIPK1 is required for survival of peripheral but not thymic gamma delta T cells." Clarification of these conclusions and seeming inconsistencies would greatly strengthen the manuscript. With respect to the actual analysis in Figure 3A, it appears that the authors used a succession of non-parametric t-tests here without any correction. It may be helpful to determine if another analysis, such as ANOVA, may be more appropriate.

Yes, we completely agree with this assessment and conclusion. While kinase dead RIPK1 does provide some rescue, this appears relatively modest, and instead supports the view, validated in figure 7, that maybe the dominant function of IKK in 𝛾𝛿 T cells is to activate NF-κB dependent survival signals. Nevertheless, RIPK1^D138N^ does provide some significant rescue, which allows some peripheral cells to repopulate and demonstrates that IKK is repressing RIPK1 mediated cell death. It is actually not trivial to assess the relative importance of IKK-RIPK1 and IKK-NF-κB functions. In the IKKΔT^CD2^ RIPK1^D138N^ mice, we prevent RIPK1 induced death, but still lack the NF-κB-dependent survival signal. Consistent with this, the ~1log reduction in 𝛾𝛿 numbers between WT and IKKΔT^CD2^ RIPK1^D138N^ mice is actually similar to what we observe in the absence of REL subunits (Fig. 7) which is a smaller reduction than we observe in IKKΔT^CD2^ mice. What would have been ideal is to have a scenario where IKK regulation of RIPK1 was defective but NF-κB survival signalling was intact. This would reveal the full impact of loosing IKK dependent regulation of RIPK1 alone, which we suspect would result in substantial cell death that could not be blocked by NF-κB. Unfortunately, we not have or know of suitable mouse mutants to test this. This is quite a nuanced discussion and we now clarify the scope and extent of conclusions we can draw (p. 7, 11).

(6) The conclusion that the alternative pathway is redundant for the development and persistence of the major gamma delta T cell subsets is at odds with a previous report demonstrating that Relb is required for gamma delta T17 development (Powolny-Budnicka, I., et al., Immunity 34: 364-374, 2011). This paper also reported the involvement of RelA in gamma delta T17 development. The present manuscript would be greatly improved by the inclusion of a discussion of these results.

Thank you - we include a discussion of these papers now (p12).

(7) The data in Figures 1C and 3A are somewhat confusing in that while both are from the lymph nodes of IKKdeltaTCD2 mice, the data appear to be quite different (In Figure 3A, the frequency of gamma delta T cells increases and there is a near complete loss of the CD27+ subset. In Figure 1A, the frequency of gamma delta T cells is drastically decreased, and there is only a slight loss of the CD27+ subset.)

Yes, we agree these do like quite different and could be confusing. The lymph nodes from IKKΔT^CD2^ lack αβ T cells and B cells, and so the cellularity is much lower than normal. Consequently, the percentage representation of remaining cells can be more noisy, while total cellularity calculations are more consistent. This is not an issue in the other strains that all have more cells in lymph nodes. We now show plots from spleen of the same mice which appear better aligned with additional splenic data shown in Figure 1.

**Reviewer #2 (Public review):**
(1) All approaches used confer changes to the entire T cell compartment. Therefore, the authors are unable to resolve whether the observations are mediated by direct and/or indirect effects (e.g., disorganized lymphoid architecture impacting maintenance/survival/homing).

We address this important point in the discussion (p11-12). The impacts of gene deletions upon αβ and 𝛾𝛿 T cells operate independently of one another (as also discussed in response to reviewer 1). For instance, the phenotype of αβ T cells is identical in IKKΔT^CD2^ and IKKΔT^CD4^ mice - 𝛾𝛿 T cells are only targeted in IKKΔT^CD2^ mice. Similarly, the phenotype of 𝛾𝛿 T cells is similar in IKKΔT^CD2^ vs Casp8.IKKΔT^CD2^ strains. αβ T cells are absent from IKKΔT^CD2^ but present in near normal numbers in Casp8.IKKΔT^CD2^ mice. Others have also noted that 𝛾𝛿 T cell development is normal in Rag deficient mice (10.1126/science.1604321). In any case, an absence of αβ T cells is expected to promote 𝛾𝛿 T cell survival in the absence of competition for common utilised cytokines such as IL-7 and IL-15, though we do not see much evidence for this in mice with and without αβ T cells such as IKKΔT^CD2^ vs Casp8. IKKΔT^CD2^ strains. We do now discuss the potential contribution of trans-conditioning for type 17 𝛾𝛿 T cell development (p12).

(2) Assessment of factors that impact T cell numbers in the periphery is necessary. Are there observable changes to the proliferation, survival, and migration of gd T cell subsets?

In IKKΔT^CD2^ and Casp8. IKKΔT^CD2^ deficient strains, we infer a defect in survival, since they lack peripheral 𝛾𝛿 T cells, despite normal thymic development. Their absence made it hard to assess proliferation and migration, though 𝛾𝛿 T cells were absent from all lymphoid organs. The conclusions that defective survival is responsible for the absence of 𝛾𝛿 T cells in the different strains is also supported by the rescue of IKKΔT^CD2^ and Casp8ΔT^CD2^ strains by kinase dead RIPK1D138N. Furthermore, the presence of small numbers of residual populations in lymph nodes and spleen of IKKΔT^CD2^ and Casp8ΔT^CD2^ strains demonstrates that migration patterns were normal. Were cells unable to recirculate, they might be expected to fail to leave the thymus, or to accumulate in the spleen. We so no evidence of either of these scenarios.

(3) TCRd chain usage, especially among type 3 gd T cells, should be assessed.

We did not unfortunately, assess chain usage, choosing rather to rely of phenotypic identity of specific subsets, which we show in figure 5C, was extremely robust. IL-17 was only secreted by CD27– CD44^hi^ 𝛾𝛿 T cells, while IFN-gamma was only secreted by CD27+ CD44^hi^ 𝛾𝛿 T cells. We argue that the production of these key effector cytokines is the most direct test of a subsets functional identity and the phenotypic designation is robust.

(4) The functional consequences of IKK signaling on gd T cells were largely unaddressed. Cytokine analyses were performed only in the RIPK1D138N Casp8∆TCD2 model, leaving open the question of how canonical NF-κB-dependent signaling impacts the long-term functionality of gd T cells.

Yes, we agree this remains an open question around the transcriptional mechanisms by which NFκB signalling promotes cell survival, and one best addressed in future studies. We did not perform cytokine staining more widely, because the cytokine assay relies on short term re-stimulation of T cells with PMA and ionomycin. PMA activates PKC which in turn activates NF-κB signalling to elicit the cytokine response measured in this assay. As such, the results of such assays would be hard to interpret. We agree it would be interesting to investigate the functional consequences of REL deficiency in future studies, although this may need a more nuanced setting where 𝛾𝛿 T cells are not lost as a result of their defective survival.

(5) The authors suggest that Caspase 8 is required for the development and maintenance of type 3 gd T cells. While the authors discussed the limitations of assessing adult mice in interpreting the data, it seems like a relatively straightforward experiment to perform.

We did attempt these experiments with collaborators by analysing type 17 𝛾𝛿 T cell development in fetal thymic organ culture (FTOC). However, the GM mice are not so easy to breed and generating the large numbers of embryos required to set up the FTOCs proved too challenging and we were unable to generate these data.

(6) While analyses of Casp8∆TCD2 RIPK1D138N mice suggest that loss of adaptive and type 1 gamma delta T cells in Casp8∆TCD2 animals is due to necroptosis, the contribution of RIPK3 kinase activity remains unexamined. RIPK3 activity determines whether cells die via necroptosis or apoptosis in RIPK1/Caspase8-dependent signaling, and inclusion of this analysis would strengthen mechanistic insights.

Given time and resources, it would have been ideal to confirm necroptotic cell death by alternative knockouts, such as RIPK3 or MLKL. However, formation of the necrosome is dependent on kinase active RIPK1, since autophosphorylation of RIPK1 changes its conformation to allow recruitment of RIPK3 and MLKL and formation of the necrosome. Therefore, the rescue of CASPASE8 deficient T cells from cell death by kinase dead RIPK1 is very solid genetic evidence of necroptosis.

(7) Canonical NF-κB signaling through cRel alone was not evaluated, leaving a gap in the understanding of transcriptional pathways required for gd T cell subsets.

This was assessed in p105/RelA knockout strain, which only express cREL. What we lacked was an assessment of what RelA/p50 dimers can support in the absence of cREL. We do however, show the impact of RelA single deficiency, and RelA/p50 deficiency.

In truth, we had many REL deficient strains and it was challenging to make all the combinations we wanted. However, we try to compensate for this by discussing what cREL:cREL dimers and cREL:P50 dimers are capable of doing by analysing 𝛾𝛿 T cell development in p105/RELA DKO and RELA KO mice - these do show that cREL:P50 can compensate in the absence of RELA, but cREL:cREL cannot.

**Reviewer #3 (Public review):**
Weaknesses:The paper would benefit greatly from a graphical abstract that could summarize the key findings, making the key findings accessible to the general immunology or biochemistry reader. Ideally, this graphic would distinguish the requirements for NF-κB signals sustaining thymic γδ T cell differentiation from peripheral maintenance, taking into account the various subsets and signaling pathways required. In addition, the authors should consider adding further literature comparing the requirements for NF-κB /necroptosis pathways in regulating other non-conventional T cell populations, such as iNKT, MAIT, or FOXP3+ Treg cells. These data might help position the requirements described here for γδ T cells compared to other subsets, with respect to homeostatic cues and transcriptional states.

Thank you - we have added such discussions. We are happy to add a graphical abstract if journal constraints permit this.

Last and least, there are multiple grammatical errors throughout the manuscript, and it would benefit from further editing. Likewise, there are some minor errors in figures (e.g., Figure 3A, add percentage for plot from IKKDT.RIPK1D138N mouse; Figure 7, “Adative").

Thank you !